# CBEC inventory optimization model design based on spatiotemporal attention and transformer architecture

Zongping Lin[1], Yingyi Huang[2]*, Jing Yang[3], Chunhu Cui[4], Yabin Lian[5], Honglei Zhang[6]

**1** School of Economics and Management, Quanzhou University of Information Engineering, Quanzhou, Fujian, China, **2** School of Business, Ningbo Tech University, Ningbo, Zhejiang, China, **3** School of Economics and Business, Xiamen City University, Quanzhou, Fujian, China, **4** Tsinghua University School of Economics and Management, Beijing, China, **5** Academy of Art & Design, Minnan Science and Technology College, Quanzhou, China, **6** Beijing Meitu Home Technology Co., Ltd, Beijing, China

* huangyingyi198205@163.com

## Abstract

To solve the problem of inaccurate long-term prediction of cross-border inventory encountered by cross-border enterprises in their experience, this paper proposes a cross-border inventory prediction model based on a spatiotemporal perception Transformer. Specifically, firstly, an improved temporal aware self-attention mechanism is adopted to mine potential temporal trends and spatial heterogeneity features in cross-border inventory, and an accurate spatiotemporal correlation matrix is established to obtain global spatiotemporal features. Secondly, we simulate the multi-level diffusion process of inventory data in the road network using multi-scale diffusion convolution, which captures the local spatial features of nodes across multiple neighborhood ranges. Finally, a multi-dimensional feature fusion module is used to adaptively fuse the captured spatiotemporal features and output prediction results. The experimental results show that compared with the ASTGNN model with the highest prediction accuracy, the method proposed in this paper performs better in MAE, MAPE, and RMSE, which are reduced by 7.6%, 4.2%, and 1.1%, respectively.

## 1. Introduction

To better meet consumer needs, merchants have to store some products in advance in warehouses on platforms such as Amazon. Still, the storage costs on these platforms are quite high, resulting in a large proportion of merchants' storage costs in their working capital. Some merchants also choose to store their products in low-cost overseas warehouses, which will increase the logistics of the goods. Compared with domestic logistics, cross-border logistics is more complex, involving multiple links such as domestic transportation, customs inspection, international transportation, and final distribution. This results in longer transportation cycles for cross-border e-commerce (CBEC) products and more difficult inventory management [1].

**Data availability statement:** https://zenodo.org/records/15803491,DOI:10.5281/zenodo.15803491.

**Funding:** This work was supported by the Xiamen Natural Science Foundation Project entitled Research on Cross-Border Marketing Strategy Transformation of Xiamen Enterprises under Multimodal AI Models (grant number 3502Z202573311 to J.Y.) and the Interim Research Findings of the 2025 Fujian Provincial Department of Science and Technology Natural Science Foundation Program (grant number 2025J08355 to Z.P.L.).

**Competing interests:** The authors have declared that no competing interests exist.

The advancement of deep learning technology in recent years has significantly accelerated research on inventory prediction in CBEC. Long Short-Term Memory (LSTM) and Gated Recurrent Unit (GRU) are commonly used methods for processing time series data, which can effectively capture complex temporal features in inventory data [2]. To address the impact of spatial information on inventory prediction, researchers introduced convolutional neural networks to extract spatial correlations between different regions. For example, Peng et al [3]. Used GRU and Graph Convolutional Network (GCN) to extract temporal and spatial features from inventory data and proposed a Time Graph Convolutional Network (TCGN) method.

In recent years, the method-based Transformer has made great progress in the research of temporal data prediction [4]. However, there are still the following problems in Transformer work:

1. The autoregressive generation method used in the Transformer model has caused an error accumulation phenomenon. The Transformer uses the teacher forcing mechanism during training, while gradually generating sequences through regression during inference. This inconsistency between training and inference leads to exposure bias, combined with the Transformer serial inference process, causing errors to gradually accumulate with increasing prediction step size. Although planned sampling can alleviate the above problems to some extent, the output of inventory prediction, which is a type of homogeneous task, is usually a continuous numerical value, which is more sensitive to errors than discrete labels (such as words). Combined with the autoregressive generation method of the Transformer, it further promotes the phenomenon of error accumulation.

2. The attention mechanism based on a single parameter cannot capture the dynamic spatiotemporal dependencies of inventory data that are time sensitive. The existing attention mechanisms often only use the same set of parameters at all times, and even if we can use time information as input features, it cannot fundamentally solve the problem of time sensitive attention.

To address these issues, this paper proposes a spatiotemporally aware Transformer model for CBEC inventory prediction. This model includes three parallel feature encoding modules for extracting spatiotemporal features. Firstly, by using the Spatial Aware Self-attention (SASA) module and the TASA module, combined with graph convolution and time-domain one-dimensional convolution and self-attention mechanisms, potential temporal trends and spatial heterogeneity features in inventory data are deeply explored to obtain accurate global spatiotemporal features. Next, by simulating the multi-level diffusion process of inventory, the multi-scale diffusion convolution (MDC) module is used to extract local spatial features. Finally, the spatiotemporal features extracted by these three feature encoding modules are input into the multivariate feature fusion module for adaptive fusion to achieve accurate inventory prediction [5].

## 2. Related works

Inventory managers predict future inventory demand at a certain point in time by analyzing a series of known information, such as historical sales records, inventory

history, and other factors that affect demand, which may be long-term or short-term [6]. There are mainly two types of methods for demand forecasting: one relies on traditional statistical methods for forecasting models, and the other uses machine learning techniques for forecasting models.

## 2.1. Prediction model based on traditional statistics

Common methods include exponential smoothing, moving average, regression prediction, autoregressive integrated moving average (ARIMA), and combination prediction model.

A power demand forecasting model that combines the exponential smoothing and ARIMA techniques was created by Nugraha [7]. The accuracy of projecting electricity demand for various nations has improved, according to an experimental study of monthly power energy time series. In their study of aircraft material demand forecasting, Zhong et al. [8] used a third-order exponential smoothing method with automatic smoothing coefficient adjustment. The findings demonstrated that this enhanced third-order exponential smoothing method can significantly increase the accuracy of aircraft material demand forecasting. Billah et al. [9] constructed a more general quadratic moving average model with two unequal step sizes, and experiments have shown that this model has better prediction accuracy than the current quadratic moving average model. Kumar et al. [10] proposed a prediction model combining the Holt-Winters index smoothing and ARIMA model, taking into account seasonality, market trends, and cyclical patterns, to solve the problem of accurate prediction of supply chain and inventory management in enterprises.

However, the exponential smoothing method is suitable for predicting large amounts of data with short cycles, but cannot handle complex factors. The moving average method is suitable for handling changes in demand cycles, but only for upward trends. The ARIMA model is suitable for short-term forecasting in relatively stable situations, but ignores the factors affecting demand. Traditional statistical methods are more suitable for linear stationary prediction. For situations with large demand fluctuations and complex influencing factors, it is necessary to combine machine learning and other methods to improve prediction accuracy.

## 2.2. Prediction model based on machine learning

Prediction model based on machine learning can be divided into two categories: one is unsupervised learning methods, among which clustering methods are one of the most commonly used unsupervised learning methods in prediction.

Sun et al. [11] considered that different traffic flow patterns could affect short-term traffic flow prediction results, and designed a combined model of K-means clustering and GRU. This model clusters historical traffic flow data, fully considering the diversity of traffic flow patterns and improving the accuracy of prediction.

Another type is supervised learning methods, which use historical data to train models and use known relationships between inputs and outputs to predict future data. An effective learning technique for symmetric extreme learning machine clusters was presented by Xing et al. [12]. It may increase the computing efficiency of the model and convert large-scale data into various challenges on small-scale data. In terms of prediction, more and more scholars are using neural network models for prediction, due to the powerful nonlinear modeling and adaptive learning capabilities of neural networks, such as recurrent neural networks (RNN), LSTM, and GCN. Due to its cyclic feedback property, RNN is a connectionist model with internal states or short-term memory, suitable for solving sequence problems such as speech classification, sequence prediction, and sequence generation [13]. To solve the problems of gradient vanishing and exploding in RNN, Joshi et al. [14] proposed a hybrid ensemble learning prediction model based on the LSTM network and applied it to e-commerce demand forecasting. Zhang et al. [15] proposed a prediction method based on the ARMA-LSTM combination model for e-commerce logistics transportation prediction problems. This method combines the advantages of both models and considers the linear and nonlinear characteristics of time series, with good experimental results. GCN is a specific implementation of graph neural networks that introduce convolution operations to aggregate node features [16]. Song et al. [17] proposed a

spatiotemporal times synchronous graph convolutional network (STSGCN) modeling model, Simultaneously capturing spatiotemporal dependencies by constructing local spatiotemporal maps.

## 3. Methodology

### 3.1. The overall framework of the proposed method

E-commerce inventory prediction can be modeled as a function that maps observed inventory data to future inventory states, such as order volume, production volume, etc. Therefore, e-commerce inventory prediction can be seen as a sequence modeling problem that constructs the relationship between historical data and future data through sliding window technology. The inventory prediction problem can be expressed as learning a function that can predict the inventory level for the next T step given the inventory level for the past T steps. This process can be described as:

$$\left[ X^{t-T+1}, X^{t-T+2}, \cdots, X^{t} \right] \rightarrow \left[ Y^{t+1}, Y^{t+2}, \cdots, Y^{t+T} \right]$$

(1)

Inventory data not only has temporal correlation between single nodes and short-term spatial correlation between multiple nodes, but also has long-term spatiotemporal dependencies among all nodes in the road network. To fully explore the spatiotemporal dependence of inventory levels, this paper proposes an inventory prediction model that integrates multiple spatiotemporal self-attention mechanisms. The overall network architecture is shown in. It mainly consists of a data embedding layer, an attention mechanism layer, and an output layer. In addition to embedding temporal information, the data embedding layer also embeds spatial information. The attention mechanism layer introduces four different self-attention mechanisms and fuses them through the feature fusion layer to identify long-term dynamic spatiotemporal correlations in inventory data, and finally predicts them through the output layer.

### 3.2. Data embedding layer

As shown in Fig 1, considering the temporal correlation of inventory, the current inventory often has a strong correlation with the order volume and production volume of the previous period, and has great similarity with the same time of the previous day and the same time of the previous week. To fuse the input data, this article combined the inventory data from the same time period last week, the inventory data from the same time period the day before, and the prior step data at the current time point. This fusion is represented by the following formula:

$$X = \left[ X^{t-T+1}, X^{t-T+2}, \cdots, X^{t} \right]$$

(2)

$$Y = \left[ Y^{t+1}, Y^{t+2}, \cdots, Y^{t+T} \right]$$

(3)

$$Z = \left[ X^{t-7\times\tau+1}, X^{t-7\times\tau+2}, \cdots, X^{t-7\times\tau+T} \right]$$

(4)

$$X_{input} = Linear(Concat(X, Y, Z))$$

(5)

Where $\tau$ represents the total number of times the detector records in a day. Transform the original input data $X$ into $X_{input} \in \mathrm{R}^{T\times N\times d}$ through a fully connected layer, where $d$ is the embedding dimension.

The road network in reality can be modeled as an irregular graph structure, which is often transformed into an analyzable structure using corresponding Laplacian matrices to capture its features and connectivity relationships. The standardized Laplacian matrix is defined as:

 

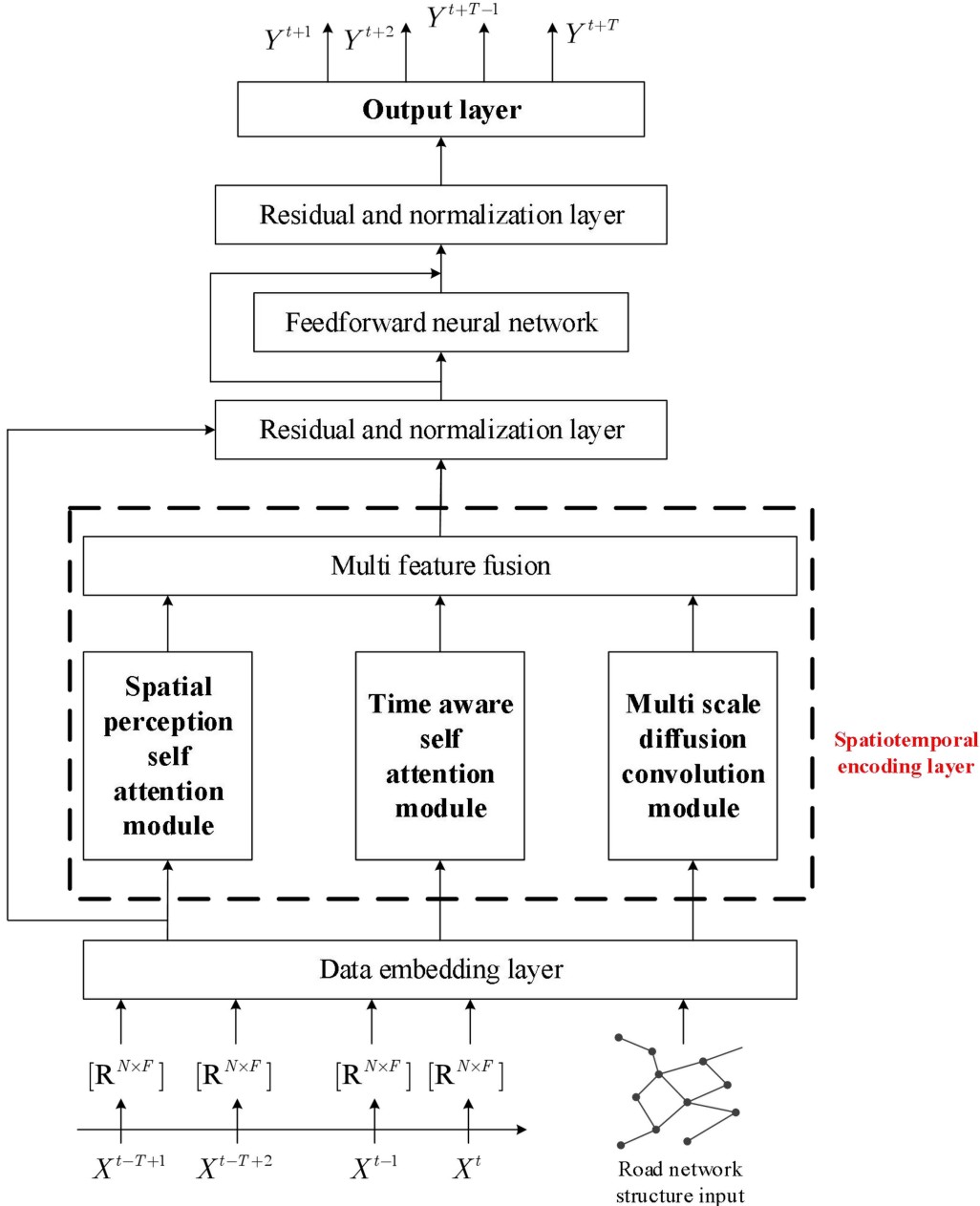

**Fig 1. The overall framework diagram of the method presented in this article.**

$$L = I_N - D^{-1/2}AD^{-1/2} = U\Lambda U^T \qquad (6)$$

Where $A \in \mathrm{R}^{N \times N}$ is the adjacency matrix, $I_N$ is the identity matrix, and the degree matrix $D \in \mathrm{R}^{N \times N}$ is a diagonal matrix composed of node degrees. The eigenvalue composition of matrix $\Lambda$ for matrix $L$ is a diagonal matrix, and $U$ is the Fourier basis. Here, linear projections on the minimum $k$ non trivial feature vectors are used to generate spatial Turalaplus embeddings, which are then extended to the input data dimension through fully connected layers, denoted as

$X_S \in \mathbf{R}^{T \times N \times d}$. Afterwards, the spatial information is encoded using the position encoding method in the traditional Transformer model, as shown:

$$X_S = \begin{cases} \sin\left(\frac{pos}{1000^{\frac{2i}{d_{model}}}}\right), i = 2k \\ \cos\left(\frac{pos}{1000^{\frac{2i}{d_{model}}}}\right), i = 2k+1 \end{cases} \tag{7}$$

Where pos represents the encoding of the pos position, and $d_{model}$ represents the encoding length. Finally, the above two embedding vectors are superimposed to obtain the output of the data embedding layer:

$$X_{data} = Linear(X_{input} + X_S) \tag{8}$$

### 3.3. Spatiotemporal encoding layer

The spatiotemporal encoding layer is the main part of the model, consisting of three main modules: SASA module, TASA module, and MDC module, used to model complex dynamic spatiotemporal correlations.

**3.3.1. Spatial perception self-attention module.** The spatial perception self-attention module is shown in the Fig 2.

Firstly, for input X, a GCN is used to generate the query matrix $Q_S$ and key matrix $K_S$, and a linear transformation is used to generate the value matrix $V_S$.

$$GCN(X_{data}) = \text{ReLU}(\tilde{D}^{-1/2}\tilde{A}\tilde{D} - 1/2X_{data}W_G) \tag{9}$$

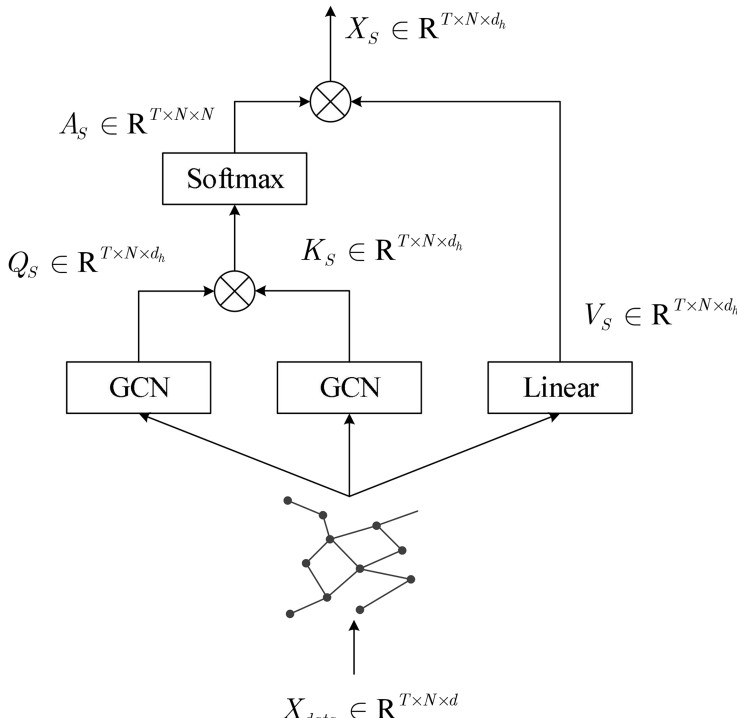

**Fig 2. Spatial perception self-attention module.**

$$Q_S = GCN(X_{data}), K_S = GCN(X_{data}), V_S = X_{data}W_S \tag{10}$$

Where $\tilde{A} = A + I$, $\tilde{D}$ is the degree matrix corresponding to $\tilde{A}$, and $W_G$ and $W_S \in R^{d \times d_h}$ are learnable parameter matrices, where $d_h$ is the feature dimension of the self-attention head. Next, perform scaled dot product attention calculation on the query matrix and key matrix to obtain the spatial attention weight matrix, and perform weighted summation on the value matrix to obtain the output of a single SASA head.

$$X_S = \text{Softmax}\left(\frac{Q_S K_S^T}{\sqrt{d_h}}\right) V_S \tag{11}$$

Where the Softmax function represents normalizing the matrix.

### 3.3.2. Time aware self-attention module.

To accurately capture relationships in time series, this model introduces a time aware self-attention mechanism. This mechanism replaces the linear transformation used to generate queries and key matrices in traditional self-attention models by adopting a one-dimensional convolutional network in the time domain. This convolutional network processes traffic data at current and adjacent time points, extracts new features through convolution operations, and enables the model to identify local change patterns in inventory data and match them with time points with similar change trends. The calculation method of time aware self-attention mechanism is as follows.

$$Q_T = \Phi_Q * X_{data}, K_T = \Phi_K * X_{data}, V_T = X_{data}W_T \tag{12}$$

$$X_T = \text{Softmax}\left(\frac{Q_T K_T^T}{\sqrt{d_h}}\right) V_T \tag{13}$$

Where $*$ represents convolution operation, and the size of the convolution kernel is $3 \times 1$. $\Phi_Q$ and $\Phi_K$ are convolution kernel parameters. $W_T \in R^{d \times d_h}$ and $X_T \in R^{N \times T \times d_h}$ are respectively the learnable parameter matrix and the output of a single time aware self-attention head.

### 3.3.3. Multi scale diffusion convolution module.

On this basis, the model uses MDC to enhance the local spatial correlation modeling of important adjacent regions of nodes, fully extracting spatial features of inventory flow data from both global and local perspectives. The k-order diffusion is used to simulate the impact of k diffusion of inventory flow along the road network on reachable nodes. The calculation formula is:

$$X_{adj}^k = C^k X_{data} W_{adj}^k \tag{14}$$

Where $W_{adj}^k \in R^{d \times d_{DC}}$ is a learnable parameterized matrix, $d_{DC}$ is the feature dimension of the diffusion convolution module, and $C = D^{-1}A$ is the transition matrix of the diffusion process. To explore the potential spatial correlation in inventory data, a learnable adaptive matrix is introduced, and its calculation method is as follows:

$$A_{adp} = \text{Softmax}(\text{ReLU}(E_1 E_2^T)) \tag{15}$$

The $k$ -order diffusion process based on adaptive matrix is defined as:

$$X_{adp}^k = C_{adp}^k X_{data} W_{adp}^k \tag{16}$$

Where $W_{adp}^k \in R^{d \times d_{DC}}$ is a learnable parameterized matrix. To obtain an integrated representation of spatial features from multiple neighborhood ranges, the original input, displayed diffusion results, and implicit diffusion results are concatenated and input into a fully connected network for adaptive fusion, resulting in the output of a MDC module:

$$X_{DC} = Concat(X_{data}, X_{adj}^1, \cdots, X_{adj}^K, X_{adp}^1, \cdots, X_{adp}^K) W_{DC} \qquad (17)$$

Where concat represents matrix concatenation operation, and $W_{DC} \in R^{(2K+1)d_{DC} \times d_{DC}}$ is a learnable parameter matrix.

The spatiotemporal encoding layer integrates the three feature encoding modules mentioned above to capture spatiotemporal features in parallel. The outputs of each module are concatenated and input into a fully connected network for adaptive fusion, resulting in the output of the spatiotemporal encoding layer:

$$X_{ST} = Concat(X_{DC}, X_S^1, \cdots, X_S^{h_S}, X_T^1, \cdots, X_T^{h_T}) W_{ST} \qquad (18)$$

Where $h_S$ and $h_T$ are the number of attention heads in the spatial and temporal perception self-attention module, and $W_{DC} \in R^{d \times d}$ is the learnable parameter matrix.

The output layer first aggregates the spatiotemporal features captured by each encoder by skipping connections, obtaining:

$$X_{out} = \sum_{i=1}^{l} X^i W_{sk}^i \qquad (19)$$

Where $X^i \in R^{T \times N \times d}$ is the output of the i-th encoder, and $W_{sk}^i \in R^{d \times d_{sk}}$ is the learnable parameter matrix. Then, a double-layer fully connected network is used for prediction, and multiple prediction results are output at once. The calculation formula is as follows:

$$Y = ReLU(ReLU(X_{out} W_O^1) W_O^2) \qquad (20)$$

Where $W_O^1 \in R^{d_{sk} \times 1}$ and $W_O^2 \in R^{T \times \tau}$ are learnable parameter matrices, and $Y$ is the final output prediction result of the model.

## 4. Experimental results

### 4.1. Experimental preparation

On a laptop computer with an Intel (R) Core (TM) i7-10510U processor, a clock speed of 1.8 GHz, and 16 GB of memory, all model training and inference experiments were conducted using the PyTorch deep learning framework within the Jupyter Notebook 6.4.6 environment.

Z is a medium-sized clothing foreign trade enterprise located in Hangzhou, Zhejiang. In recent years, it has sold overseas to Southeast Asia through CBEC platforms such as Shopee, with customers mainly located in countries such as Thailand, Cambodia, Vietnam, and Myanmar. Consider adopting two models: overseas warehouse and border warehouse. Among them, overseas warehouses are close to customer points and have fast order response times, but there are problems such as high costs and difficulty in handling unsold goods; On the contrary, the rental cost of border warehouses is low, but there are problems such as high order response costs. Z Company relies on both border warehouses and overseas warehouses to build its supply chain network, and transports through dedicated land routes. The experimental dataset spans from January 1, 2022, to December 31, 2023. We employed a fixed-origin split for evaluation, with the training set covering January 1, 2022, to September 30, 2023, the validation set from October 1, 2023, to November

30, 2023, and the test set comprising the final month, December 1 to December 31, 2023. The model's input window length was set to 28 days, meaning it uses the preceding 28 days of data to forecast a prediction horizon (T) of 7 days. The covariates used include historical inventory levels, order volume, production volume, and temporal features (day of the week and holiday indicators). For preprocessing, minimal missing values (<0.5%) were handled via linear interpolation, and all numerical features were normalized to the [0, 1] range using Min-Max scaling fitted solely on the training data to prevent leakage.

In this article, the model optimizer uses Adam W, the learning rate scheduler uses CosineLRScheduler, the initial learning rate is set to 0.001, the batch size is set to 16, the number of model iterations is set to 100, and the remaining parameters in the model structure are shown in Table 1.

## 4.2. Baseline methods and evaluation indicators

Three indicators—mean absolute error (MAE), root mean square error (RMSE), and mean absolute percentage error (MAPE)—were chosen for the experiment To assess the model's performance and examine the forecast outcomes.

To ensure a fair and consistent comparison, all baseline models and the proposed model were implemented using the same feature sets, input window length (28 days), and prediction horizon (7 days). The implementations of the baseline models were based on their official open-source repositories: GWNET [18], STSGCN [17], and T-GCN [19] were sourced from their respective authors' GitHub pages. Hyperparameter tuning for all models was conducted via a grid search over key parameters: learningrate $\in \{0.001, 0.0005\}$, hiddenunits $\in \{32, 64\}$, and numberoflayers $\in \{2, 3\}$. Training was regulated by an early-stopping criterion with a patience of 10 epochs, monitoring the validation loss to avoid overfitting and ensure convergence.

Due to real-time updates, there are some missing values and outliers in the data. Therefore, before conducting model training, Huber Loss is used as the loss function. This function uses absolute error when the error is large, which can enhance the robustness of the model to outliers and noise. Using continuously differentiable squared error when the error is small can help accelerate the convergence of the loss function. The Huber Loss function is shown below:

$$\text{Huber Loss} = \begin{cases} \frac{1}{N} \sum_{i=1}^{N} \frac{1}{2}\left(Y_i - \tilde{Y}_i\right)^2, |Y_i - \tilde{Y}| \le \delta \\ \frac{1}{N} \sum_{i=1}^{N} \left(\delta|Y_i - \tilde{Y}| - \frac{1}{2}\delta^2\right), |Y_i - \tilde{Y}| > \delta \end{cases}$$

(21)

Where $\delta$ represents the threshold for switching the control loss function, which is set to 2 in this article. To verify the effectiveness of using Huber Loss to solve the problem of excessive sensitivity of mean square error to outliers, this paper uses mean square error (MSE) and Huber Loss as loss functions to train the model, and the training results are shown in Fig 3.

**Table 1. Model structure parameters.**

| Parameter | Value |
| --- | --- |
| Number of spatiotemporal encoding blocks | 6 |
| Diffusion step $k$ | 2 |
| Dimension of spatiotemporal embedding layer features | 64 |
| Multi scale diffusion convolution module feature dimension | 16 |
| Feature dimension of spatial perception self-attention module | 32 |
| Time aware self-attention module feature dimension | 16 |
| Self-attention head feature dimension | 8 |
| Output layer skips connection feature dimension $d$ | 256 |

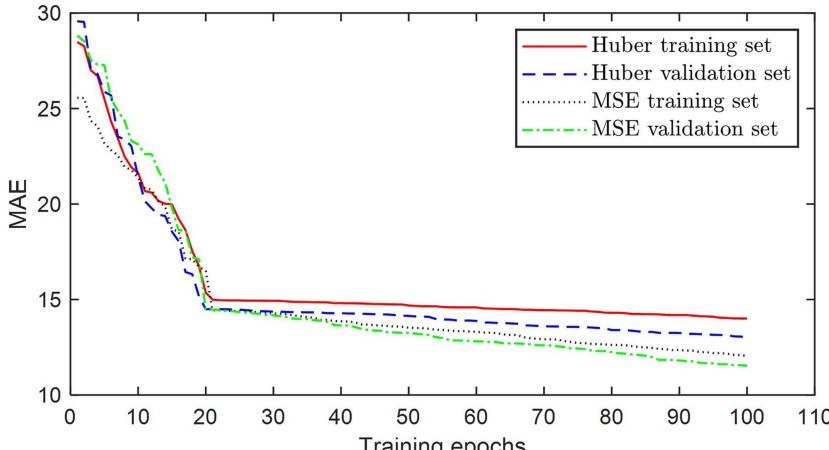

**Fig 3. The error reduction curve of the training set and validation set during the model training process.**

Fig 3 shows the decrease curves of MAE in the training and validation sets during model training using different loss functions. It can be seen that Huber Loss has better convergence performance than MSE on both the training and validation sets, and has a greater advantage on the validation set. This is because when using MSE as the loss function, the model overfits outliers and noise in the data, resulting in a decrease in its generalization ability on the validation set.

## 4.3. Experimental results

### 4.3.1. Experimental comparison results.
The experimental results of the method proposed in this article and the comparative model are shown in Table 2.

The error indicators of the method proposed in this article are superior to all comparison models. Compared with ASTGNN, which has the highest prediction accuracy, MAE, MAPE, and RMSE are reduced by 7.6%, 4.2%, and 1.1%, respectively.

This article takes clothing, food, home appliances, and, cosmetics in CBEC as inventory verification objects. The inventory prediction results of the method proposed in this article are shown in Fig 4. In Fig 4, it can be seen that the method proposed in this paper has good prediction results for inventory in the next year.

Next, we will analyze the inventory level. Assuming that we need to ensure that the inventory is around $0.5 \times 10^5$, we need to determine the current purchase quantity based on current orders and future inventory forecasts. The inventory levels for different methods are shown in.

From Fig 5, it can be seen that the inventory level of the method proposed in this article will remain around $0.5 \times 10^5$ for the next 50 days, satisfying the maximization of revenue, while other methods have significant fluctuations. Once it exceeds $0.5 \times 10^5$, it indicates the need for more costs, while below $0.5 \times 10^5$, it will reduce trading volume and thus lower profits.

**Table 2. Performance comparison of different models.**

|  | T-GCN | STSGCN | GWNET | GMAN | SSTAT | ASTGNN | Proposed |
|---|---|---|---|---|---|---|---|
| MAE(%) | 20.91 | 21.67 | 19.09 | 19.14 | 19.24 | 18.81 | 18.05 |
| MAPE(%) | 14.80 | 14.37 | 13.34 | 13.19 | 12.83 | 12.47 | 12.04 |
| RMSE(%) | 33.38 | 34.39 | 31.73 | 31.60 | 30.97 | 30.76 | 29.66 |

**Fig 4. The prediction results of the method proposed in this article on different products.**

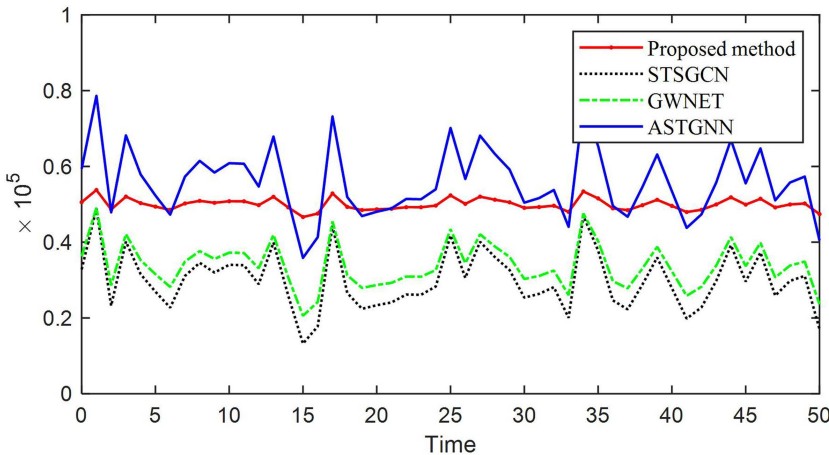

**Fig 5. Inventory quantity for the next 50 days.**

**4.3.2. Ablation experiment.** Spatiotemporal coding layer ablation experiment: To verify the effectiveness of each module in the spatiotemporal encoding layer, this paper replaces or removes some components based on the proposed method, forming the following variant model:

1. w/o GCN: Remove MDC;

2. w/oMDC: Replace MDC with ordinary graph convolution;

3. w/oTASA: Replace time aware self-attention mechanism with traditional time self-attention mechanism;

4. w/oSASA: Replace spatial perceptual self-attention mechanism with traditional spatial self-attention mechanism.

The experimental results are shown in Table 3.

**Table 3. Performance comparison of different models.**

|         | w/oGCN | w/oMDC | w/oTASA | w/oSASA | Proposed |
|---------|--------|--------|---------|---------|----------|
| MAE(%)  | 19.81  | 18.91  | 18.32   | 18.23   | 18.05    |
| MAPE(%) | 13.27  | 12.87  | 12.25   | 12.24   | 12.04    |
| RMSE(%) | 32.12  | 31.20  | 30.21   | 29.81   | 29.66    |

It can be seen that the algorithm proposed in this article outperforms all variant models in three indicators. This indicates that the global and local fusion spatial modeling method adopted in this article can effectively improve the prediction accuracy of the model. The results of the ablation experiment indicate that the model structure design proposed in this paper is reasonable, and the improvements of each module have independent contributions to the prediction performance.

Data embedding layer ablation experiment. To verify the effectiveness of the spatiotemporal embedding method, this paper designs the following variant models:

1. w/oSPE: Remove spatial position embeddings;

2. w/oTPE: Remove time and position embeddings;

3. w/o Period: Remove time period embeddings.

The ablation experiment results of the above variants are shown in Table 4. Removing any type of spatiotemporal embedding will cause the model to lose necessary spatiotemporal features, thereby affecting the prediction accuracy.

**4.3.3. Model complexity analysis.** The results of running different models are shown in Table 5. In this article, both GMAN and ASTGNN adopt an encoder decoder architecture, and GMAN uses global attention in both spatial and temporal dimensions, resulting in the highest training time and parameter count. ASTGNN adopts autoregressive prediction method, which only outputs the prediction results of a single time step at a time, resulting in low inference efficiency.

## 4.4. Limitations

Although the proposed model demonstrates superior performance in CBEC inventory prediction, several limitations remain to be addressed in future work.

First, the computational complexity of the self-attention mechanism remains a challenge, especially when dealing with large-scale spatial networks or long-term historical data. Although our model achieves a balance between performance and efficiency compared to other Transformer-based baselines, the quadratic complexity of self-attention with respect to sequence length may still hinder its application in real-time scenarios or on edge devices with limited computational resources. Future work will explore more efficient attention mechanisms, such as linear attention or sparse attention, to reduce computational overhead without significantly sacrificing accuracy.

Second, the current model primarily relies on historical inventory data and basic temporal features. While these features capture major trends, they do not fully incorporate external factors that may significantly impact inventory demand,

**Table 4. Performance comparison of different models.**

|         | w/oSPE | w/oTPE | w/oPeriod | Proposed |
|---------|--------|--------|-----------|----------|
| MAE(%)  | 18.95  | 18.62  | 18.48     | 18.05    |
| MAPE(%) | 12.98  | 12.66  | 12.57     | 12.04    |
| RMSE(%) | 33.21  | 31.98  | 30.12     | 29.66    |

**Table 5. Performance comparison of different models.**

|  | Training time (s) | Reasoning time (s) | Parameter quantity | MAE |
|---|---|---|---|---|
| GMAN | 289.3 | 22.7 | $9 \times 10^5$ | 15.87 |
| ASTGNN | 149.2 | 51.7 | $6.2 \times 10^5$ | 14.29 |
| Proposed | 112.8 | 8.3 | $2.6 \times 10^5$ | 12.22 |

such as sudden market fluctuations, promotional activities, macroeconomic policies, or unexpected events. In the future, we plan to enhance the model by integrating multi-source data, including social media trends, search engine indexes, and economic indicators, to improve the robustness and adaptability of predictions under volatile conditions.

Third, the model's performance is dependent on the quality and granularity of the data. In practice, cross-border inventory data often suffer from missing values, noise, and inconsistent recording frequencies across different regions or platforms. Although we used Huber Loss and interpolation to mitigate some of these issues, a more sophisticated data imputation or denoising module could be beneficial. Future research could explore joint modeling of data reconstruction and prediction within an end-to-end framework, or leverage generative model to synthesize realistic training data under sparse observations.

Lastly, the current model is designed and validated primarily for inventory prediction in the fashion and consumer goods sectors. Its effectiveness in other industries with different demand patterns remains to be verified. We intend to extend the evaluation to more diverse product categories and explore domain adaptation techniques to improve the generalizability of the model across different industrial contexts.

## 5. Conclusion

This article proposes a CBEC inventory prediction model based on spatiotemporal perception Transformer by improving the traditional self-attention mechanism. By combining time-domain one-dimensional convolution and graph convolution with self-attention mechanism, the spatiotemporal perception self-attention module can fully consider the influence of temporal trend and spatial heterogeneity on spatiotemporal relationships, and capture accurate global spatiotemporal features. In addition, the model adopts a global local spatial modeling method. Based on the capture of global spatial features by the SASA module, a MDC module is used to enhance the extraction of local spatial features of important adjacent regions of nodes. Finally, the captured multi-dimensional spatiotemporal features are adaptively fused to generate prediction results. Compared with other algorithms, the algorithm proposed in this article has higher prediction accuracy. In the next step of our work, we will continue to optimize the computational cost of the model and investigate the impact of other factors on inventory, as well as explore other methods for mining dynamic spatiotemporal correlations.

## Acknowledgments

We thank the anonymous reviewers whose comments and suggestions helped to improve the manuscript.

## Author contributions

**Conceptualization:** Zongping Lin, Yingyi Huang.

**Data curation:** Zongping Lin.

**Formal analysis:** Zongping Lin, Yingyi Huang, Honglei Zhang.

**Investigation:** Yingyi Huang, Jing Yang, Chunhu Cui, Yabin Lian.

**Methodology:** Yingyi Huang, Jing Yang.

**Project administration:** Yabin Lian.

**Resources:** Zongping Lin, Jing Yang.

**Software:** Honglei Zhang.

**Supervision:** Jing Yang, Chunhu Cui, Honglei Zhang.

**Validation:** Chunhu Cui, Yabin Lian.

**Visualization:** Chunhu Cui.

**Writing – original draft:** Zongping Lin, Yingyi Huang, Jing Yang, Yabin Lian.

**Writing – review & editing:** Chunhu Cui, Honglei Zhang.

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
