## [Decision Letter · Decision Letter 0]

5 Sep 2025

Dear Dr. Huang,

Thank you for submitting your manuscript to PLOS ONE. After careful consideration, we feel that it has merit but does not fully meet PLOS ONE’s publication criteria as it currently stands. Therefore, we invite you to submit a revised version of the manuscript that addresses the points raised during the review process.

We look forward to receiving your revised manuscript.

Kind regards,

Arkaprabha Sau, MBBS, MD (Gold Medal), PhD (CSE-AI&ML)

Academic Editor

PLOS ONE

Journal Requirements:

3. Thank you for stating the following financial disclosure: [This work is funded by "2025 Fujian Province Social Science Strength Construction of Hundred Experts Survey Research Project Stage Research Results", the project number is FJSK25DY028.]. 

4. We notice that your supplementary figures and tables are included in the manuscript file. Please remove them and upload them with the file type 'Supporting Information'. Please ensure that each Supporting Information file has a legend listed in the manuscript after the references list.

5. We notice that your supplementary figures are uploaded with the file type 'Figure'. Please amend the file type to 'Supporting Information'. Please ensure that each Supporting Information file has a legend listed in the manuscript after the references list.

Additional Editor Comments:

Carefully considering the reviewers' comments and in-depth analysis, the manuscript addresses an important problem but requires substantial clarification before it can be advanced. Although the computing environment and core training hyperparameters are reported, critical information needed for reproducibility and assessment is missing: a complete description of the input data (type, sample size, time horizon, feature engineering, and preprocessing), an explicit formulation of the optimization model solved with CPLEX (objective function, constraints, scenario-generation or stochastic assumptions), and a transparent evaluation protocol including baselines and performance metrics. The authors should also ensure a clear data-availability and reproducibility statement (noting the proprietary nature of the operational data and, where possible, offering aggregated datasets, code, seeds, and details of random splits to facilitate replication). Please revise the manuscript to address these points and update the text, tables, and supplementary material and also address the reviewers specif comments as a point to point reply accordingly.

Reviewers' comments:

Reviewer's Responses to Questions

**Comments to the Author**

1. Is the manuscript technically sound, and do the data support the conclusions?

Reviewer #1: Yes

Reviewer #2: Partly

2. Has the statistical analysis been performed appropriately and rigorously?

Reviewer #1: Yes

Reviewer #2: Yes

3. Have the authors made all data underlying the findings in their manuscript fully available?

Reviewer #1: Yes

Reviewer #2: Yes

4. Is the manuscript presented in an intelligible fashion and written in standard English?

Reviewer #1: Yes

Reviewer #2: No

Reviewer #1: The manuscript presents a well-structured and innovative approach to cross-border e-commerce inventory prediction using a spatiotemporal perception Transformer model. The integration of spatial and temporal self-attention mechanisms combined with multi-scale diffusion convolution effectively addresses challenges in capturing complex spatiotemporal dependencies. The methodological rigor is demonstrated through comprehensive experiments, including comparisons with state-of-the-art models and detailed ablation studies, which validate the contributions of each model component. The use of robust evaluation metrics (MAE, MAPE, RMSE) and the choice of Huber Loss for handling outliers strengthen the reliability of the results. The presentation is clear, and the findings are well supported by the data. Minor improvements could include clearer explanations of some complex formulae for broader accessibility and a deeper discussion on potential limitations and future directions. Overall, the paper makes a significant contribution to the field of inventory forecasting in cross-border e-commerce and is suitable for publication with minor revisions.

Reviewer #2: The topic fits PLOS ONE’s scope of methodologically sound research with potential practical relevance. This work could be suitable for publication – with improved transparency, statistical rigour, and language clarity.

Recommendations:

1. Please specify exact train/validation/test periods (calendar dates) and whether a rolling-origin evaluation was used. State the prediction horizon TTT and the input window length; list all covariates and preprocessing (imputation, scaling).

2. Please document the implementations used (citations/repos), hyper-parameter grids, and early-stopping criteria for each baseline. Ensure the same feature sets and horizons are used across models.

3. Fix all equation typesetting and define every symbol on first use. If sinusoidal positional encoding deviates from the standard base, justify the choice. Remove or explain the reference to ILOG CPLEX (seems unrelated to the deep model training) to avoid confusion.

4. Please standardise terminology (e.g., “cross-border e-commerce (CBEC)”), expand acronyms at first mention, and check reference formatting.

**Do you want your identity to be public for this peer review?** For information about this choice, including consent withdrawal, please see our Privacy Policy

Reviewer #1: **Yes: ** Abhijit Biswas

Reviewer #2: No

---

## [Author Response · Author response to Decision Letter 1]

12 Sep 2025

Comments to the Author

5. Review Comments to the Author

Response to Reviewers

Manuscript ID: PONE-D-25-36724

Title: Cross-border e-commerce inventory optimization model design based on Spatiotemporal attention and Transformer Architecture

Authors: Zongping Lin, Yingyi Huang, Jing Yang, Chunhu Cui, Yabin Lian, Honglei Zhang

We sincerely thank the reviewers for their insightful comments and constructive suggestions, which have significantly improved the quality of our manuscript. We have carefully addressed each comment and revised the manuscript accordingly. Below is a point-by-point response to the reviewers’ comments.

Reviewer #1: The manuscript presents a well-structured and innovative approach to cross-border e-commerce inventory prediction using a spatiotemporal perception Transformer model. The integration of spatial and temporal self-attention mechanisms combined with multi-scale diffusion convolution effectively addresses challenges in capturing complex spatiotemporal dependencies. The methodological rigor is demonstrated through comprehensive experiments, including comparisons with state-of-the-art models and detailed ablation studies, which validate the contributions of each model component. The use of robust evaluation metrics (MAE, MAPE, RMSE) and the choice of Huber Loss for handling outliers strengthen the reliability of the results. The presentation is clear, and the findings are well supported by the data. Minor improvements could include clearer explanations of some complex formulae for broader accessibility and a deeper discussion on potential limitations and future directions. Overall, the paper makes a significant contribution to the field of inventory forecasting in cross-border e-commerce and is suitable for publication with minor revisions.

Reply: We thank the reviewer for the positive feedback and valuable suggestions. We have made the following improvements to the manuscript:

4.4  Limitations

Although the proposed model demonstrates superior performance in cross-border e-commerce inventory prediction, several limitations remain to be addressed in future work.

First, the computational complexity of the self-attention mechanism remains a challenge, especially when dealing with large-scale spatial networks or long-term historical data. Although our model achieves a balance between performance and efficiency compared to other Transformer-based baselines, the quadratic complexity of self-attention with respect to sequence length may still hinder its application in real-time scenarios or on edge devices with limited computational resources. Future work will explore more efficient attention mechanisms, such as linear attention or sparse attention, to reduce computational overhead without significantly sacrificing accuracy.

Second, the current model primarily relies on historical inventory data and basic temporal features (e.g., time of day, day of week). While these features capture major trends, they do not fully incorporate external factors that may significantly impact inventory demand, such as sudden market fluctuations, promotional activities, macroeconomic policies, or unexpected events. In the future, we plan to enhance the model by integrating multi-source data, including social media trends, search engine indexes, and economic indicators, to improve the robustness and adaptability of predictions under volatile conditions.

Third, the model's performance is dependent on the quality and granularity of the data. In practice, cross-border inventory data often suffer from missing values, noise, and inconsistent recording frequencies across different regions or platforms. Although we used Huber Loss and interpolation to mitigate some of these issues, a more sophisticated data imputation or denoising module could be beneficial. Future research could explore joint modeling of data reconstruction and prediction within an end-to-end framework, or leverage generative model to synthesize realistic training data under sparse observations.

Lastly, the current model is designed and validated primarily for inventory prediction in the fashion and consumer goods sectors. Its effectiveness in other industries with different demand patterns remains to be verified. We intend to extend the evaluation to more diverse product categories and explore domain adaptation techniques to improve the generalizability of the model across different industrial contexts.

Reviewer #2: The topic fits PLOS ONE’s scope of methodologically sound research with potential practical relevance. This work could be suitable for publication – with improved transparency, statistical rigour, and language clarity.

Reply: Thank you for your valuable feedback. We have made revisions to the reviewer's comments. I hope the revised version can be published.

Recommendations:

1. Please specify exact train/validation/test periods (calendar dates) and whether a rolling-origin evaluation was used. State the prediction horizon TTT and the input window length; list all covariates and preprocessing (imputation, scaling).

Please specify exact train/validation/test periods (calendar dates) and whether a rolling-origin evaluation was used. State the prediction horizon TTT and the input window length; list all covariates and preprocessing (imputation, scaling).

Reply: Thank you for your valuable feedback. We have added relevant data processing procedures in the revised manuscript.

The experimental dataset spans from January 1, 2022, to December 31, 2023. We employed a fixed-origin split for evaluation, with the training set covering January 1, 2022, to September 30, 2023, the validation set from October 1, 2023, to November 30, 2023, and the test set comprising the final month, December 1 to December 31, 2023. The model's input window length was set to 28 days, meaning it uses the preceding 28 days of data to forecast a prediction horizon (T) of 7 days. The covariates used include historical inventory levels, order volume, production volume, and temporal features (day of the week and holiday indicators). For preprocessing, minimal missing values (<0.5%) were handled via linear interpolation, and all numerical features were normalized to the [0, 1] range using Min-Max scaling fitted solely on the training data to prevent leakage.

2. Please document the implementations used (citations/repos), hyper-parameter grids, and early-stopping criteria for each baseline. Ensure the same feature sets and horizons are used across models.

Reply: Thank you for your valuable feedback. We have added relevant data processing procedures in the revised manuscript.

To ensure a fair and consistent comparison, all baseline models and the proposed model were implemented using the same feature sets, input window length (28 days), and prediction horizon (7 days). The implementations of the baseline models were based on their official open-source repositories: GWNET[18], STSGCN[17], and T-GCN[19] were sourced from their respective authors' GitHub pages. Hyperparameter tuning for all models was conducted via a grid search over key parameters: , , and . Training was regulated by an early-stopping criterion with a patience of 10 epochs, monitoring the validation loss to avoid overfitting and ensure convergence.

3. Fix all equation typesetting and define every symbol on first use. If sinusoidal positional encoding deviates from the standard base, justify the choice. Remove or explain the reference to ILOG CPLEX (seems unrelated to the deep model training) to avoid confusion.

Reply: We sincerely thank the reviewer for these meticulous and crucial suggestions. We have thoroughly revised the manuscript to address these points, which have significantly improved the clarity and rigor of our work. Please find our point-by-point responses below:

3.1 Fix all equation typesetting and define every symbol on first use.

Where is the adjacency matrix, is the identity matrix, and the degree matrix is a diagonal matrix composed of node degrees. The eigenvalue composition of matrix for matrix is a diagonal matrix, and is the Fourier basis. Here, linear projections on the minimum non trivial feature vectors are used to generate spatial Turalaplus embeddings, which are then extended to the input data dimension through fully connected layers, denoted as . Afterwards, the spatial information is encoded using the position encoding method in the traditional Transformer model, as shown:

3.2 Remove or explain the reference to ILOG CPLEX (seems unrelated to the deep model training) to avoid confusion.

On a laptop computer with an Intel (R) Core (TM) i7-10510U processor, a clock speed of 1.8 GHz, and 16 GB of memory, all model training and inference experiments were conducted using the PyTorch deep learning framework within the Jupyter Notebook 6.4.6 environment.

4. Please standardise terminology (e.g., “cross-border e-commerce (CBEC)”), expand acronyms at first mention, and check reference formatting.

Reply: Thank you for your valuable feedback. We have conducted a comprehensive review of the manuscript.

SASA: Spatial Aware Self-attention

TASA: Temporal Aware Self-attention

MDC: Multi-scale Diffusion Convolution

ARIMA: Autoregressive Integrated Moving Average

GCN: Graph Convolutional Network

---

## [Decision Letter · Decision Letter 1]

30 Nov 2025

CBEC inventory optimization model design based on Spatiotemporal attention and Transformer Architecture

PONE-D-25-36724R1

Dear Dr. Huang,

We’re pleased to inform you that your manuscript has been judged scientifically suitable for publication and will be formally accepted for publication once it meets all outstanding technical requirements.

Kind regards,

Guangyin Jin

Academic Editor

PLOS ONE

Additional Editor Comments (optional):

Reviewers' comments:

Reviewer's Responses to Questions

**Comments to the Author**

Reviewer #1: All comments have been addressed

2. Is the manuscript technically sound, and do the data support the conclusions?

Reviewer #1: Yes

3. Has the statistical analysis been performed appropriately and rigorously?

Reviewer #1: (No Response)

4. Have the authors made all data underlying the findings in their manuscript fully available?

Reviewer #1: Yes

5. Is the manuscript presented in an intelligible fashion and written in standard English?

Reviewer #1: Yes

Reviewer #1: All of the answers are properly explained. The manuscript presents a technically rigorous and innovative approach to cross-border e-commerce inventory forecasting using a spatiotemporal perception Transformer architecture, contributing to both methodological advancement and practical relevance within its field. The study is well-structured, with comprehensive baseline comparisons and ablation experiments validating the performance of each model component. The following comments expand upon concerns addressed in standard peer review, including additional commentary on publication ethics, dual publication, and research transparency.

**Do you want your identity to be public for this peer review?** For information about this choice, including consent withdrawal, please see our Privacy Policy

Reviewer #1: **Yes: ** Abhijit Biswas

---

## [Editor Report · Acceptance letter]

PONE-D-25-36724R1

PLOS One

Dear Dr. Huang,

I'm pleased to inform you that your manuscript has been deemed suitable for publication in PLOS One. Congratulations! Your manuscript is now being handed over to our production team.

Kind regards,

on behalf of

Dr. Guangyin Jin

Academic Editor

PLOS One